# Histological Patterns and Mammographic Presentation of Invasive Lobular Carcinoma Show No Obvious Associations

**DOI:** 10.3390/cancers16091640

**Published:** 2024-04-24

**Authors:** Gábor Cserni, Rita Bori, Éva Ambrózay, Orsolya Serfőző

**Affiliations:** 1Department of Pathology, Bács-Kiskun County Teaching Hospital, Nyíri út 38., 6000 Kecskemét, Hungary; rita.bori@gmail.com; 2Department of Pathology, Albert Szent-Györgyi Faculty of Medicine, University of Szeged, Állomás u. 1., 6725 Szeged, Hungary; 3Breast Diagnostic Department of MaMMa Zrt at Kecskemét, Bács-Kiskun County Teaching Hospital, Nyíri út 38., 6000 Kecskemét, Hungary; ambrozayster@gmail.com (É.A.); serfike@gmail.com (O.S.); 4MaMMa Egészségügyi Zrt. Center, Kapás u. 22., 1027 Budapest, Hungary

**Keywords:** invasive lobular carcinoma, breast cancer, mammography, histological pattern, multifocality

## Abstract

**Simple Summary:**

Breast cancer is a heterogenous disease from many aspects. The most common special-type, invasive lobular carcinoma, is not homogenous either. It may have several mammographic appearances, some of which make screen detection difficult. Histologically, lobular carcinoma has a peculiar morphology consisting of non-cohesive cells often arranged in single file, but this classical morphology may often be admixed with less easily recognizable histological patterns, such as the solid, the alveolar, the trabecular, or the tubule-forming, some of which are characterized by the regained cohesion of the tumor cells. Whether the dominant histological patterns are associated with specific mammographic presentation is not known, but such associations may be important from diagnostic aspects looking at radio-pathological correlations. It was believed that the classical histologic pattern might be more often occult, whereas more cohesive or dense patterns might be more often mass forming. Our analysis failed to demonstrate obvious associations between radiological and pathological presentations.

**Abstract:**

Invasive lobular carcinoma of the breast has different mammographic appearances, including spiculated or lobulated masses, architectural distortion, increased breast density, and the possibility of also being occult. Histologically, the morphology is also variable, as several patterns have been described beside the classical one, including the solid, the alveolar, the trabecular, the one with tubular elements, and others. Of 146 ILC cases, 141 were reviewed for mammographic appearance and 136 for histological patterns by two radiologist and two pathologists, respectively; 132 common cases were analyzed for possible associations between mammographic presentation and the histological patterns. Interobserver agreement on the presence or absence of a given mammographic morphology ranged from 45% (increased density) to 95% (occult lesion); the most common radiomorphology was that of a spiculated mass. Interobserver agreement on the presence or absence of a given histological pattern ranged between 79% (solid) and 99% (classical) but was worse when semi-quantification was also included. The mammography–pathology correlation was less than optimal. Multifocality was more commonly detected by histology. The identification of a mammographic mass lesion often coincided with a mass-like lesion on the histological slides and vice versa, but nearly half of the mammographically occult lesions were felt to have masses on histological slides assessed grossly. Histological patterns showed no obvious associations with one or the other mammographic appearance.

## 1. Introduction

Invasive lobular carcinoma (ILC) is a common, special type of breast cancer that is represented in the World Health Organization (WHO) classification of breast tumors since the first edition in 1968, where it received one paragraph with four lines and a black and white picture probably illustrating it [1]. With subsequent editions [2,3,4], the description grew step by step, and by the time of the latest edition, the fifth one in 2019 [5], it was 4.5 pages in extent, reflecting the knowledge gathered during the 51 years separating the two publications. Today, the WHO defines ILC as an invasive carcinoma “composed of discohesive cells that are most often individually dispersed or arranged in a single-file linear pattern” [5]. Morphological patterns recognized by the WHO classification include the solid, the alveolar, the pleomorphic and the tubulolobular one, and a mixed group, in which the classical pattern is seen associated with one or several other patterns [5]. Besides those listed in the WHO classification and reproduced here, many other patterns have been described, as nicely summarized by Christgen et al., including the trabecular, the plexiform, the solid papillary, the signet-ring cell variants, and those with extracellular mucin production or having tubular elements [6]. These morphological variants deviate from the single-file presentation or the dispersed isolated cell infiltrative patterns of classical ILCs, or they show a cellular component that deviates from the cytology seen in the classical manifestation.

Solid ILC is characterized by sheets of discohesive cells and increased mitotic activity [5]. Alveolar ILC comprises globoid structures with 20 or more cells [5]. The pleomorphic variant is defined on the basis of tumor cell size being greater than 4-fold the size of small lymphocytes [5]. Finally, the tubulolobular variant shows a mixture of tubules and uniform small cells infiltrating in a linear pattern [5]. It is worth noting that many pathologists, including us, do not think that the tubulolobular carcinoma described by Fisher et al. [7] represents a variant of ILC [6], as it is characteristically positive for E-cadherin immunohistochemistry [8,9,10], and better represents a relatively well-differentiated non-lobular carcinoma, as reflected by its three-dimensional reconstruction [11]. In contrast, there is a variant of ILC, where tubules are formed and retain their E-cadherin negativity; these are referred to as ILC with tubular elements, and an E-cadherin–P-cadherin switch has been proposed as the molecular mechanism in the background of this cohesive morphology in CDH1-mutant, E-cadherin-negative ILCs [12]. The trabecular pattern is characterized by trabeculae, and as described by Martinez and Azzopardi [13], this can refer to both single-cell-thick and thicker trabeculae, of which the first is practically identical with the single filing of the classical pattern, but thicker trabeculae will be used here to define the trabecular pattern of infiltration [6]. Whereas the classical pattern of ILC is easy to recognize, some morphological variants give rise to interpretational problems and are a potential source of erroneous classification, as highlighted in a recent reproducibility study [14]. The routine use of E-cadherin immunohistochemistry can reduce uncertainty in classification [15].

As concerns radiological presentation of ILCs, even the latest edition of the blue book [5] highlights architectural distortion as a common and typical mammographic manifestation of the tumor. This is a well-known manifestation and makes the recognition of the tumor rather difficult [16]. However, we are aware of the possibility of ILC remaining mammographically occult and of the fact that it has the ability to form masses. In fact, spiculated masses may be the most common mammographic presentation of ILCs according to previous studies analyzing this issue [17,18,19]. On the basis of focality, mass formation, or its lack, Tot defined a diffuse infiltrative pattern beside unifocal and multifocal tumors, which is characterized by the lack of a mass and infiltration reminiscent of a spiderweb [20]. These tumors are associated with a poor outcome independent of immunohistochemical biomarkers of prognosis [21].

It may be hypothesized that classical pattern ILCs are less commonly mass forming and more commonly occult; whereas histological patterns make the recognition of ILC more difficult because of non-lobular (“ductal”) features, such as tubule formation, a more aggregated cell population might be more often mass forming.

In the present study, we looked at the mammographic and pathological appearance of ILCs, the reproducibility of the classification of these appearances, and their possible correlations.

## 2. Materials and Methods

ILC cases diagnosed at the Department of Pathology, Bács-Kiskun County Teaching Hospital, between July 2016 (the introduction of digital mammography) and December 2023 were extracted from the recorded histopathological diagnoses of breast cancer. Cases with mixed ILC (i.e., ILC and non-ILC invasive carcinomas) were excluded, as well as cases first seen in our breast imaging unit after primary systemic treatment or presenting with recurrent disease.

Two radiologists (ÉA, OS) involved in reporting screening mammography, retrospectively reviewed the mammographic images of the cases and independently categorized the cases as having either architectural distortion, an increase in density, circumscribed/lobulated mass, spiculated mass, or no radiological features (occult) (Figure 1); multiple categories could also be opted for the same mammogram. During the analysis, a hierarchical approach was also used: if a mass lesion was partly spiculated, partly lobulated, or circumscribed, it was considered spiculated; whenever there was a mass lesion, this was ranked first, and architectural distortion or increased density were considered only if there was no mass lesion in addition. Comments could be added, for example, if microcalcifications were present. The observers also recorded whether the lesion looked multifocal on mammography—MULT(MG). The period encompassed in the study involved digital mammography with a Giotto 3DL (until February 2021) or a Giotto Class (from February 2021) unit (both from IMS Giotto SpA, Sasso Marconi, Italy). From September 2021, tomosynthesis was also performed on 31 cases and reviewed within the frames of this study, making the morphological evaluation more delicate. The list selected on the basis of the pathology diagnoses included a few cases diagnosed elsewhere, and these had to be excluded from the radiology review. ILCs with discrepant individual categorization were then re-analyzed mutually to reach consensus on morphological features.

Histological slides were also recovered from the archives of the pathology department, and the slides were independently analyzed by two pathologists (BR, CG) reporting breast cancers routinely. The morphological patterns (classical, solid, tubule-forming, trabecular, alveolar, pleomorphic, and others specified as ILCs) were recorded (Figure 2), as well as the presence of multifocality or its absence. Because tubule formation and other non-classical patterns of ILCs reduce the reproducibility of the diagnosis of ILC [14], it was decided to quantitate the presence of the patterns similarly to the tubule scoring of the grading scheme of breast cancers: 3+: >75% presence, 2+: >10% to 75% presence, 1+: up to 10% presence, and 0: absence. When looking for interobserver agreement, besides this four-tiered classification on the presence or absence of a given pattern, we also looked for the substantial presence (>10%) vs. absence or minor presence (0–10%) of a given pattern considering the 10% cut-off value that is also used in histological typing; i.e., per convention, less than 10% components are not considered and do not result in the diagnosis of mixed histological types. Additionally, any presence of a pattern versus its absence was also analyzed (0% vs. >0%).

Multifocality was approached in two ways: Tot defined invasive breast cancers as unifocal, multifocal, or diffuse in distribution [22,23] and multifocality requiring only the foci to be separated by tissue without invasive cancer, i.e., non-contiguous foci at any distance fulfilled the criterion of multifocality—MULT(TOT). This classification proved to be of prognostic significance, with multifocal tumors having worse prognosis than unifocal ones [22,23]. The International Collaboration on Cancer Reporting (ICCR) suggests that invasive carcinomas with (satellite) foci within 5 mm distance should be measured as one tumor focus, and stemming from this pragmatic approach, the second evaluation of multifocality required a distance of greater than 5 mm between foci to fulfill the criterion: MULT(ICCR) [24]. Multifocality in this review was assessed on individual slides, and could not identify multifocality that was represented by different foci on different slides. Therefore, it was decided at the beginning that whenever a discrepancy between mammographic multifocality and its lack on histology is encountered, this will prompt the integration of the grossing description and better clarification of this feature with a combined pathological macro- and micromorphology. Furthermore, looking at the tumor-containing slides grossly and under the microscope, the pathologists registered whether there was a clearly identifiable mass or not. Masses that were not malignant (e.g., concomitant fibroadenoma, papilloma) were also noted but were separately marked from malignant masses. This information was also integrated in the evaluation.

After both pathologists evaluated the slides independently, they re-analyzed all cases with discrepant interpretations and reached consensus on the originally discrepant features.

Besides descriptive statistics, the intraclass correlation coefficient (two-way random effects, absolute agreement, single rater/measurement; ICC(2,1)) was also used to reflect interrater agreement on different aspects evaluated in the study [25]. These were calculated with Excel on the basis of Real Statistics Using Excel [https://real-statistics.com/reliability/interrater-reliability/intraclass-correlation/ version Rel 8.9.1 (released 2 October 2023)] (last accessed 15 March on 2024). The interpretation of the ICC values was based on the value itself but also considered the 95% confidence intervals; <0.50, 0.5–0.749, 0.75–0.9, and >0.9 values reflected poor, moderate, good, and excellent agreement, respectively [25]. Comparisons of pattern distribution and mammographic appearances were made with the Fisher exact test (two-tailed, with *p* < 0.05 for significance), using VassarStats [Richard Lowry, http://vassarstats.net/, last accessed on 21 April 2024)]

This non-interventional study was approved by the Human Investigation Review Board of the University of Szeged (115/2019 SZTE).

## 3. Results

Due to the nature of the evaluations, there were different numbers of tumors evaluated by the radiologists and the pathologists.

Of the 146 cases of pure ILCs, 4 happened to be diagnosed elsewhere and had no remaining imaging records in our setting; these were kept for the histopathology analysis but had to be excluded from the radiological review. A further exclusion from the radiological assessment involved a single case with a 2 mm-large ILC in the setting of larger LCIS and an intraductal papilloma. Therefore, 141 ILCs were evaluated for radiologic manifestation.

Some of the ILCs seen by the radiologists were treated with neoadjuvant therapies, and because of associations with any signs of regression, the cases were excluded from the pathology review. On the other hand, cases lacking imaging data for the radiology review because the diagnostics were carried out elsewhere had their histology slides available and were included in the analysis. With these considerations, 136 cases were evaluated for the patterns of ILC.

Finally, of the 141 ILCs categorized by the radiologists and the 136 ILCs categorized by the pathologist, 132 were common and could be analyzed for radio-pathologic correlations.

All patients were females. The main clinicopathological features of the involved patients and tumors are shown in Appendix A.

### 3.1. Mammography

The absolute agreement of the two opinions on the whole of mammographic morphology was seen in 33/141 cases (23.4%). Agreement on the presence or absence of a given morphologic appearance (architectural distortion, increase in density, spiculated mass, circumscribed or lobulated mass, occult lesion) was obviously better, ranging from 45 to 95% (Table 1); however, the ICC values suggested poor or poor-to-moderate agreement, with the exception of occult lesions, where agreement (especially on the non-occult classification) was 95%. When architectural distortion and increased density were lumped together, the rate of agreement of the two radiologists was 46.8% (66/141), with greater agreement on the presence than on the absence (49 vs. 17); the ICC(2,1) of 0.14 (95%CI: −0.04–0.31) suggests poor agreement. For mass lesions lumped together independently of their contours, agreement was seen in 68.1% (96/141; 56 on presence, 40 on absence), and the ICC(2,1) of 0.42 (95%CI: 0.19–0.60) reflects poor-to-moderate agreement.

As concerns multifocality, although the agreement between the observers was 89% (125/141, only 9/125 for the presence of multifocal lesions), the ICC(2,1) reflected poor agreement (Table 1).

The consensus classification of originally divergently categorized lesions resulted in numbers between the extremes of the two radiological opinions; i.e., the consensus classification generally resulted in a decrease in cases in the given category for the observer who “overcalled” lesions to that category and in an increase in cases for the observer who “undercalled” them to the same category (Table 2). The consensus classifications were used for evaluating radio-pathological correlations.

### 3.2. Pathology

A total of 1066 normal size histological slides and 183 (7.5 × 5 cm) megaslides were reviewed.

Complete agreement on the absence or presence of the five major patterns investigated (classical, solid, alveolar, tubule-forming, trabecular) and their semiquantitative proportions per individual cases was seen in only 30/136 cases (22.1%). Agreement for the four-tiered categorization of patterns ranged between 64% and 89% depending on individual patterns (Table 3).

Agreement on the presence or absence of given patterns was seen in 79.4 to 98.5% of the cases depending on the pattern, being highest for the classical and lowest for the solid (Table 3). When up to 10% representation of a given pattern was opposed to >10% presence (such as in the case of scoring tubule formation during the histological grading of breast cancer), agreement on this two-tiered classification per pattern ranged between 85.3% and 97.1% (Table 3). Complete agreement for both presence or absence and the semiquantitative amount for individual cases was 72/136 (52.9%) in this latter setting (>10% or less) and 85/136 (63.2%) in the former (0% or >0%).

The individual and consensus classifications are reflected in Table 4. It appears that the classical and trabecular patterns have been the most commonly recognized patterns of ILC in this series, and on the opposite side, the presence of tubular elements in ILC was the least common pattern. On the basis of the agreed consensus patterns, a pure classical pattern of ILC was diagnosed in only 3/136 cases (2%), whereas dominant (>75%) classical ILC with negligible (up to 10%) non-classical pattern(s) was recorded in a further 20 cases (15%). Altogether 41 ILCs were classified as having a >75% classical pattern (Table 4). In contrast, based on the consensus interpretation, an isolated pure non-classical pattern was diagnosed in only one case, a solid ILC. Additionally, 22 cases (16%) had a major non-classical pattern with a minimal (up to 10%) classical pattern; these included 5 cases of trabecular, 2 cases of solid, and 1 case of alveolar ILC with >75% representation of the given pattern, but all of these were admixed with the minor presence of some other patterns besides the presence of minimal classical ILC. The remainder of the 22 cases had at most 75% but generally less representation of different non-classical patterns mixed. Altogether, 45 ILCs were interpreted as having a major (>75%) non-classical pattern: 5, 9, and 1 with >75% solid, trabecular, and alveolar patterns, respectively; the remaining cases with mixed patterns with no single pattern being >75% (Table 4).

The pleomorphic pattern of ILC was also recorded in eight cases at least as a focal feature of the examined tumor, but it was not part of the patterns collected for radio-pathological correlations. Although all of these pleomorphic cases had >10% classical pattern ILC, none had >75% of this pattern, and all had >10% non-classical patterns, with three cases having >75% composed of a mixture of solid, trabecular, and alveolar; of trabecular, tubule-forming, and solid; and of trabecular, alveolar, and tubule-forming patterns.

As concerns multifocality, naturally, this was seen more commonly with the Tot definition requiring just a minimum of normal tissue or in situ carcinoma between distinct foci: two-thirds of the cases (67%) were classified as having more than one focus (Table 5). When the ICCR rule for measurement was used as a basis for separating multifocal tumors, they became less frequent (22%) but were still more frequent than by mammography (Table 2). On the basis of the slides examined, there seemed to be a mass lesion in the majority of the cases (Table 5).

Some form of lobular neoplasia was seen in association with 114/136 (84%) of the cases, 6 of which were also associated with ductal carcinoma in situ; 2 further cases of this latter lesion were seen associated with ILC but without lobular neoplasia being identified. Altogether 20 cases had neither lobular neoplasia nor ductal carcinoma in situ identified.

### 3.3. Radio-Pathological Correlations

Using the consensus opinions of radiologists and pathologists involved in the study, the following statements can be made.

Of the 106 ILCs that the pathologists felt to be mass forming on the basis of a gross and microscopic examination of the slides, 81 (76%) were classified as mass forming (79 spiculated, 2 lobulated) by the radiologists. Interestingly, four of the nine mammographically occult lesions were also felt to be mass forming on the slides (three of them unanimously classified as such, and one on the basis of consensus classification). The remaining 21 cases belonged to the architectural distortion/increase in density-type presentation on mammography. Of the 87 mass-forming mammographic abnormalities considered in the dual analysis, 6 (7%) were lacking a mass on the histological slides, where of the 9 occult lesions, only 5 were felt to have no mass on the basis of the slides. For an approach of the diffuse distribution of ILCs, we have looked for cases that formed no masses on the mammogram, on the slides, or both (Table 6). Although many cases could not be classified as being either of predominant classical or non-classical (generally a mixture of different) patterns, the classifiable cases suggested that the lack of mass lesions on the histological slides or on the histological slides and the mammogram was associated with the classical histological pattern.

For multifocality, in 19 of the MULT(MG), only 4 (21%) were classified as MULT(ICCR) and all 19 as MULT(TOT). A revision of the gross finding suggested that two further cases (6, i.e., 32% altogether) could be classified as MULT(ICCR), as distant foci were blocked separately, and this multifocality could not be evidenced on individual slides. Three further gross descriptions mentioned multifocality, but the foci were not distant enough to fit the MULT(ICCR) group. Finally, eight tumors were grossly described as unifocal (only five were visible, and three were only palpable) and two as diffuse lesions (both without naked eye identifiability). In contrast, of the 29 MULT(ICCR) tumors, only 4 (14%) were identified as MULT(MG); and of 89 MULT(TOT), 19 (21%) were MULT(MG).

As concerns the histological patterns and their relation to mammographic presentation, these are summarized in Table 7. Only 82/132 cases (62%) could be assigned a histological pattern on the basis of >75% representation. Although the proportion of the distribution of spiculated mass lesions was 2:3 in favor of dominant non-classical patterns and the reverse was true for both the architectural distortion/increased density and the occult presentations, where the ratio was 2:3 in favor of the classical pattern, there was no statistically significant difference in the distributions (Fisher exact test, *p* = 0.35).

No obvious association was detected between lesions occult on mammography and their histological patterns. Eight of the nine cases had a >10% classical pattern, and one was a pure classical pattern ILC. Seven of the nine cases had a >10% non-classical pattern, of which the trabecular one predominated, but the solid and the alveolar patterns were also present in this amount in one case each. The circumscribed/lobulated mass category in its pure form was rare and was reported in only two cases, one solid and the other classical in pattern (Table 7). Spiculated mass lesions and ILCs with an architectural distortion or density increase in breast parenchyma were seen in both classical and non-classical patterns; there was no statistically significant association even if the two radio-morphologies with less than 10 cases (occult and lobulated/circumscribed mass) were excluded (Table 7, Fisher exact test, *p* = 0.14). A mass lesion (spiculated or lobulated/circumscribed) on mammography was 78% sensitive and 77% specific for a mass on the histological slides as inspected with the naked eye or under the microscope.

## 4. Discussion

ILC is sometimes difficult to identify on mammography, and it is widely acknowledged that this type of cancer may typically present as architectural distortion [5,16], may reach a relatively large size, and not uncommonly may become palpable before it is detected by imaging. A review of previous mammograms in such cases may disclose subtle and barely perceptible changes that may be compatible with the presence of smaller-size, earlier-stage ILC [16,26]. A survey among 366 radiologists, members of the Society of Breast Imaging, revealed that only 25% were confident in detecting ILC on screening mammography if the breasts were dense, and this changed to 67% in non-dense breasts; therefore, most radiologists felt that imaging beyond mammography was needed to detect ILC [27]. This is widely acknowledged even by recommendations for further imaging when the diagnosis of ILC is established, and the Hungarian National guidelines are also compatible with this [28,29]. In the present series, we studied only cases that had been identified and operated on; therefore, it is not the detection of the lesion that was assessed but its perception as mammographic alteration.

Classification into architectural distortion, increased density, masses with spiculated or better circumscribed borders, and mammographically occult lesions was far from well reproducible. Only 33/141 cases (23.4%) reached complete agreement on the presence or absence of all possible morphologies; likewise, the ICC suggested poor-to-moderate agreement. However, when individual opinions on the presence or absence of each of these morphologies were assessed separately for the 5 × 141 classifications, the agreement was 514/705 (72.9%), which looks much better. The 191/705 instances of discrepant classification required simultaneous review and consensus on classification to allow the assessment of any associations between histopathological and mammographic morphologies. Although the two-observer setting reflects the double reading approach prescribed for mammography screening, it does not allow for a proper evaluation of interobserver reproducibility. As a limitation of the present study, these agreement data reflect only the given two observers, but their degree of reported agreement can lie anywhere on the minimum and maximum scale that can be determined with ONEST (Observers Needed to Evaluate Subjective Tests) analyses using more observers [30,31,32,33]. The data must therefore be interpreted with this note of caution.

Screen-film mammography studies report somewhat different distributions between the ILC radio-morphologies. A Swedish report from Malmö suggested that 72/137 (53%) ILCs presented as spiculated opacities, and therefore, this was the predominant mammographic presentation, with the remaining cases being distributed as architectural distortion (22, 16%), occult (22, 16%), poorly defined opacities (10, 7%), or parenchymal asymmetry (5, 4%) [17]. Of the 52 ILCs seen at the Department of Radiology, University of Michigan Medical Center, Ann Arbor, 4 (8%) were occult, the majority presented as spiculated mass (33, 63%), and the remaining were encountered as asymmetrical densities (7, 13%), architectural distortions (5, 10%), microcalcifications (2, 4%), or a well-circumscribed mass (1, 2%) [18]. In a Japanese series of 61 ILCs [19], a spiculated mass was identified in 23 (38%), architectural distortion in 10 (16%), and an occult lesion in 1 (2%) case. The remaining ILCs were classified as an obscured mass (14; 23%), an indistinct mass (3; 5%), or asymmetric opacity (10; 16%) lumped together in our increased density category. A smaller Turkish series reported on 38 ILCs [34]; mass and/or architectural distortion were the most common findings in 16 (42%) cases, whereas asymmetrical density and occult lesions were encountered in 11 and 11 (29–29%) cases, respectively.

Our results are in line with the quoted reports. On the basis of the consensus classification, the majority, i.e., 91/141 (65%), of lesions were seen as spiculated masses, with 12 of these having better circumscribed areas, too; altogether, 93 mass lesions (66%) were recognized, with the remaining 38 (27%) being architectural distortion or increased densities without distinct mass lesions; such lesions were classified as different opacities or asymmetries by others [17,19]. To note, 73 further cases with architectural distortion and/or a density increase also had mass lesions detected, and in the hierarchical classification, they were categorized as mass lesions. Ten cases had no obvious mammographic signs that could be associated with ILCs, i.e., these were occult on mammography. Although microcalcifications were seen in seven cases from the whole series, these were not considered manifestations of ILC.

Besides the classical pattern of ILC, several variants have been described (for a review, see Christgen et al. [6]), some of which are recognized by the WHO, whereas others are not. We have chosen patterns that are related to infiltration, as this was felt to be more influential on the mammographic presentation. We distinguished between the classical and the trabecular [13] patterns on the basis of single versus multiple cell thickness of the cell files in the stroma and also looked for solid (sheet-like) and alveolar (spherical-to-ovoid groups of at least 20 cells) patterns. On the other hand, we did not include the plexiform morphology, where trabeculae or single files do not run in parallel but merge at different angles; these were placed mostly into the trabecular pattern, though specific comments mentioned the plexiform nature at times. The tubulolobular variant, as described by Fisher [7] and analyzed in further studies [8,9,10,11], was not included, as we felt there was more evidence in favor of this variant being a non-lobular carcinoma than it being a pattern of ILC [6]. In contrast, we identified and specifically looked for any tubule formation in ILC, as ILC with E-cadherin-negative tubular elements have been recognized as an ILC where E-cadherin is replaced by P-cadherin following a molecular switch [12]. This morphology is a potential pitfall, like the trabecular (and probably the alveolar) pattern in recognizing some ILCs as ILC, if E-cadherin is not used to assist the proper diagnosis [14,15]. Other rare patterns, such as the one with extracellular mucin production [35] or the solid papillary one [6], were also not looked for specifically but were on occasion mentioned as possible minimal representation in comments.

Knowledge of the different patterns of ILC has its importance in making the diagnosis of ILC even when these patterns are present, and this is where E-cadherin immunohistochemistry or even a CDH-1 mutational analysis can be of help [14,15].

It also seems that the different patterns have different outcomes. Probably, the first study to look at this is one from 1982; it found that the solid one had the worst outcome [36]. Of the 103 cases of ILC, 96% had agreement on classification by three pathologists into four categories: classical, solid, alveolar, or mixed, with 80% proportion required to be one of the non-mixed variants, although the mixed variant also included cases that had significant nuclear pleomorphism and would therefore qualify for either nuclear grade 3 ILCs or pleomorphic ILCs today. The distribution of the cases between the subtypes was 31, 23, 19, and 30 for the classical, solid, alveolar, and mixed variants, which is divergent from our cases, with a >75% presence of a specific pattern (Table 4); indeed, only 5% and 1% of the cases belonged in the solid and alveolar patterns, as the remaining 45 cases with a >75% non-classical pattern belonged either in the trabecular pattern (not specifically distinguished by Dixon et al. or lumped into the mixed group [36]) or was of the mixed non-classical type, and the remaining 67% of all cases were also mixed, with the classical pattern being one of the components. Certainly, the cases were diagnosed on HE stained slides only, as the knowledge about the lack of E-cadherin function behind ILC morphology was missing at the time when the data were collected and analyzed, but the examples depicted in this publication from 1982 [36] are typical of the variants illustrated. (For comparison, 78/136 cases (57%) had E-cadherin immunohistochemistry with or without immunohistochemistry for beta-catenin and/or p120 catenin in the present series). With these restrictions, the authors found that nearly half of the cases with solid and mixed ILCs were in stages III and IV, whereas nearly 80% of the two other types were in stages I and II. The survival curves of the solid (worst) and mixed ILCs were worse than those of the classical (best) and alveolar variants; no multivariate analysis was performed.

Although the classification rules in 1982 cast some doubt about the validity of the data [36], a more recent study with multivariate analysis also supported the worse prognosis related to the solid variant of ILC. With 981 cases, this is the largest ever reported single institutional series on variants of ILC: the majority were of the classical type (541, 56%) followed by alveolar (146, 15%), non-classical mixed (145, 15%), solid (104, 11%), and trabecular (38, 4%) variants [37]. However, the mixed variant included patterns such as the pleomorphic, the signet-ring cell, the histiocytoid, and the apocrine with a minimum of 50% representation for the given classification. Therefore, the classification cannot be fully converted to the one used in the present study. With a similar classification, Desmedt et al. reported decreasing proportions of classical (197, 48%), alveolar (66, 16%), solid (65, 16%), mixed non-classical (57, 14%), and trabecular (28, 7%) variants in a series of 413 ILCs [38]. In contrast to these results stands the distribution of different subtypes in a recent French series of 453 ILCs, with the classical being the most frequent (312, 68%), followed by the mixed (119, 26%), the alveolar, and the solid (11, 4% each) [39]. These low rates of solid and alveolar ILCs and the high rate of mixed patterns is more in keeping with what we saw.

We also noted that the distinction between one pattern or the other was not sharp enough to allow for good interobserver agreement. Although the agreement on the presence or absence of a given pattern ranged between 79% (for solid) and 99% (for classical), the quantitative estimations were less favorable. Many pathologists would consider the trabecular pattern as a part of the classical because single files are often admixed with files of 2–3 cells in thickness, corresponding to trabeculae. This is further supported by the lack of a representation of this pattern in the current WHO classification of breast tumors [5]. As noted by Martinez and Azzopardi, the trabecular pattern is not usually described in ILC, but this is a common pattern worthy of wider recognition [13]. Another problematic issue is the size of the alveolar structures in the alveolar variant. Based on the original description by Martinez and Azzopardi [13], a minimum of 20 non-cohesive cells and a globoid arrangement is the requirement, but then, how are smaller alveolar structures with 10–15 cells classified? An example in Figure 2 of the paper by Iorfida et al. shows an alveolar ILC with globular aggregates of “more or less than 20 neoplastic cells” [37]. Therefore, one option would be to label these as alveolar structures, provided that at least some fulfill the criterion of at least 20 cells, but another way would be to call these clumps (terminology defined by Martinez and Azzopardi) [13] mini-trabeculae of 2-3-4 cell thickness but short. The more ovoid a nearly spherical alveolar structure is, the more it approaches a short trabecula. The solid pattern is characterized by sheet-like arrangements of the cells with at most minimal stroma, but what size does a sheet need to be in order to be recognized as a minor solid component in an ILC? These are practical issues that are not addressed in current classification guidelines [5,40].

Both the trabecular- and the tubule-forming components of ILCs are recognized as features impacting the diagnosis of ILCs, as these are “ductal” features seen in no special-type invasive breast carcinomas. Although a tubule is easy to recognize, sometimes, tubules are scanty and have a closed or nearly closed lumen [13], making their interpretation questionable. Small numbers of tubules and virtually no lumen formation were features identified as causes of discrepant classifications in the present study, but in the differential diagnosis, even apoptotic cell drop-out and adipocytes surrounded by ILC cells were considered potential features that could be misinterpreted as tubular elements (Figure 3).

Personal experience would have suggested that occult cancers and those with architectural distortion and density changes would belong to the classical type of ILC, but the data did not substantiate this preliminary hypothesis. Similarly, it was felt that solid ILCs may have a more circumscribed morphology with mass formation. However, the data pointed to the majority of predominantly solid ILCs to manifest as spiculated masses, which was the most common mammographic presentation of all ILCs, although the only pure solid ILC was a circumscribed mass. No obvious association was found between more aggregated (non-classical) histological patterns and mass lesions on mammography.

Although ILCs are more often multifocal than ductal carcinomas, this was relatively rarely seen on the mammograms, justifying the recommendations of magnetic resonance imaging or the use of additional techniques to detect multifocality. Even the interobserver agreement on the mammographic categorization as multifocal vs. not was suboptimal, with an ICC(2,1) value suggesting poor agreement. In contrast, multifocality with the definition by Tot was very commonly recognized on the histological slides, and the interobserver agreement was slightly better on the basis of ICC(2,1), as it was moderate. Features reducing interrater agreement included the small size of the microscopic foci or minimal distance between foci (Figure 4). Neither of these would be expected to be readily detectable with imaging. In contrast, a multifocality requiring at least 5 mm distance between foci, MULT(ICCR), that was hypothesized to be better reflective of mammographic multifocality was found similarly reproducible but failed to be present in about two-thirds of the cases reported as multifocal on mammograms.

Diffuse cases with the worst prognosis and a lack of tumoral masses [21,41] seemed to be associated with the classical pattern of infiltration, although fewer cases with a predominant non-classical histological pattern also showed no masses on the slides (and mammography).

In summary, although architectural distortion, increased density, and mammographically occult lesions are well-known features of ILCs, our results, in keeping with those of others, suggest that the most common manifestation is that of a spiculated mass lesion. The distinction of these manifestations is not always straightforward, especially if multiple labelling is allowed; the ICC(2,1) values suggested poor-to-moderate interobserver agreement for one or the other mammographic label.

The variable histological architectural patterns of ILCs recognized beside the classical one are more “ductal”-like, especially at low power, and therefore, they are a source of potential misclassification as non-ILC. Although these patterns are described in classifications or guidelines [5,40], their descriptions do not generally provide size or proportion criteria for allowing for the diagnosis of a component in a mixed-pattern case or a pure non-classical pattern. Likewise, the agreement between observers was moderate-to-good according to the ICC(2,1) values. It is important to note that the majority of ILCs will have at least a minor component of classical ILC that might help in establishing the proper histological type of breast cancer, and alternately, most ILC will be of a mixed type with at least minor components of non-classical patterns. Although the pleomorphic pattern of ILC is defined as a cancer, with the infiltration features being typical of ILC but the cells being pleomorphic [5], our data suggest that many pleomorphic ILCs are mixed according to infiltration patterns, and non-classical patterns may be more prevalent among them than classical ones, i.e., the definition is to be amended.

MULT(MG) was better reflected by MULT(TOT), but many of the numerous cases with MULT(TOT) were not labeled as MULT(MG). The mammography–pathology correlation was therefore suboptimal. The identification of a mammographic mass lesion often coincided with a mass-like lesion on the histological slides and vice versa, but nearly half of the mammographically occult lesions were felt to have masses on histological slides assessed grossly. A lack of mass lesions (a diffuse distribution) on the slides was more common in classical ILCs. Histological patterns showed no obvious associations with one or the other mammographic appearance.

## 5. Conclusions

Despite commonly stressed mammographic presentations of ILCs (architectural distortion, increased density, occult), spiculated mass lesions were the most common presentation, and the reproducible categorization into these mammographic presentations was not perfect. The recognition of different histological patterns of ILCs also showed only moderate-to-good interobserver agreement, and while most ILCs will have a mixture of patterns, the majority will have at least a minor component of the classical one. Although mammographic masses often coincided with a mass-like lesion on the histological slides and vice versa, the radio-pathologic correlation of multifocality was less than optimal; a lack of masses was more common among classical ILCs. Histological patterns showed no obvious associations with given mammographic appearances.

## Figures and Tables

**Figure 1 cancers-16-01640-f001:**
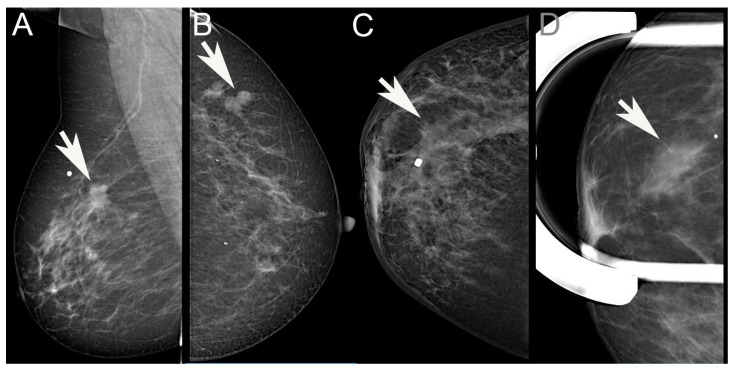
Examples of visible mammographic alterations evaluated in the study. (**A**): Spiculated mass; (**B**): Lobulated/circumscribed mass; (**C**): Architectural distortion; (**D**): Increased density. (Arrows point to the lesions. White dots label the cutaneous area where the patient palpated the lesions).

**Figure 2 cancers-16-01640-f002:**
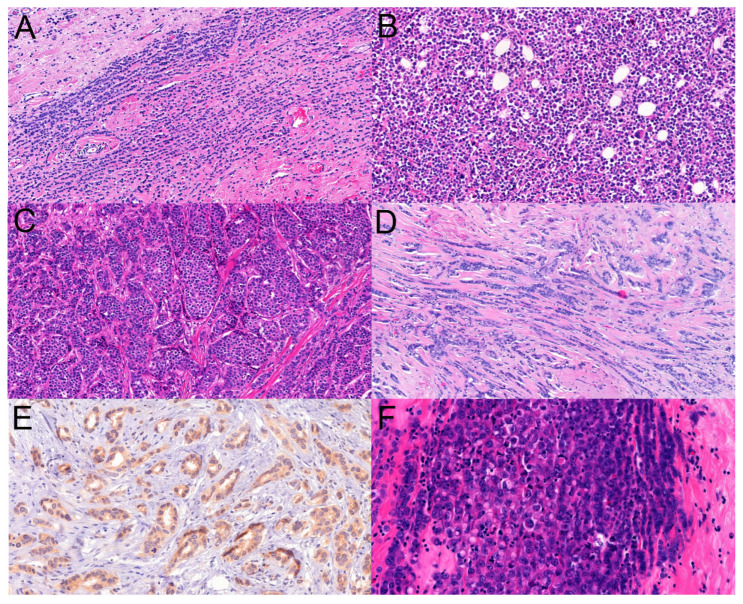
Examples of the histopathological infiltration patterns of lobular carcinomas evaluated in the study: (**A**): classical with single cell files (HE ×20); (**B**): solid (HE ×20); (**C**): alveolar (HE ×20); (**D**): trabecular (HE ×40); (**E**): ILC with tubular elements with cytoplasmic p120 immunostaining (p120 ×30), (**F**): pleomorphic (HE ×40).

**Figure 3 cancers-16-01640-f003:**
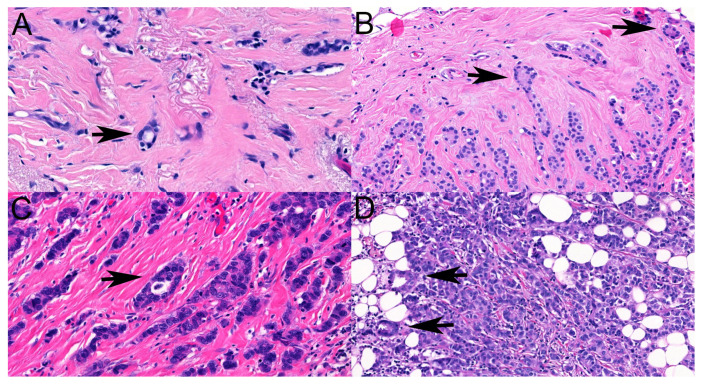
Features that can be (mis)interpreted as tubular elements (arrows): (**A**): intracytoplasmic vacuole of a single cell imitating tubule formation (HE ×63), (**B**): closed tubules with no lumen but oriented polarized cells (HE ×30), (**C**): apoptotic cell drop-out (HE ×20), (**D**): adipocytes surrounded by ILC cells (HE ×30).

**Figure 4 cancers-16-01640-f004:**
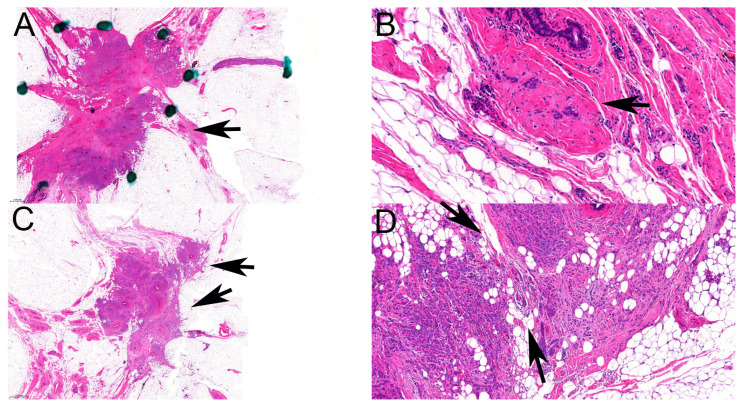
Multifocal ILC according to Tot. (**A**,**B**): Very small independent focus. (**A**): One mass-forming lesion dotted, in keeping with a spiculated morphology; an extension of lobular neoplasia at 3 o’clock position also dotted, with the site of the second focus (arrow) (HE, ×1); (**B**): Four groups of tumor cells forming the minute second focus (arrow), which are easy to miss next to the larger one (HE, ×20). (**C**,**D**): Minimal breast parenchyma between two foci. (**C**): Smaller focus (horizontal arrow) close to larger focus (arrow) (HE, ×1.2). (**D**): Adipocytes and two ducts without tumor cells (between the arrows) separating the two close foci (HE, ×10).

**Table 1 cancers-16-01640-t001:** Interrater agreement on mammographic presentation of lobular carcinomas.

	Architectural Distortion	Increased Density	Spiculated Mass	Lobulated or Circumscribed Mass	Occult	Multifocal
No agreement	42/141 (29.8)	77/141 (54.6)	42/141 (29.8)	23/141 (16.3)	7/141 (5.0)	16/141 (11.3)
Agreement	99/141 (70.2)	64/141 (45.4)	99/141 (70.2)	118/141 (83.7)	134/141 (95.0)	125/141 (88.7)
Agreement on presence	41 (41.4)	9 (14.1)	56 (56.6)	6 (5.1)	6 (4.5)	9 (7.2)
Agreement on absence	58 (58.6)	55 (85.9)	43 (43.4)	112 (94.9)	128 (95.5)	116 (92.8)
ICC(2,1)	0.42	0.07	0.41	0.29	0.60	0.37
95% CI of ICC	0.26–0.55	−0.05–0.20	0.17–0.59	0.14–0.43	0.49–0.70	0.21–0.50
ICC interpretation	poor	poor	poor	poor	moderate	poor

CI: confidence interval, ICC: intraclass correlation coefficient. Numbers in parentheses represent percentages of all cases in data rows 1 and 2 and of all cases with agreement in data rows 3 and 4.

**Table 2 cancers-16-01640-t002:** Individual and consensus mammographic classification of the ILC cases.

	Architectural Distortion	Increased Density	Architectural Distortion or Increased Density	Spiculated Mass	Lobulated or Circumscribed Mass	Mass	Occult	Multifocal
Observer 1	49	11	55	96	23	100	9	24
Observer 2	75	81	121	55	14	61	7	9
**Consensus**	**69**	**54**	**101**	**91**	**14**	**93**	**10**	**19**
Hierarchical; Observer 1	na	na	31	96	4	100	9	na
Hierarchical; Observer 2	na	na	72	55	6	61	7	na
**Hierarchical; Consensus**	**na**	**na**	**38**	**91**	**2**	**93**	**10**	**na**

na: not applicable.

**Table 3 cancers-16-01640-t003:** Agreements in the classification of given histological patterns of ILC in the series (n = 136).

	Classical	Non-Classical (Together)	Solid	Tubular Elements	Trabecular	Alveolar
No agreement	32 (23.5%)	35 (25.7%)	48 (35.3%)	15 (11.0%)	37 (27.2%)	49 (36.0%
Agreement	104 (76.5%)	101 (74.3%)	88 (64.7%)	121 (89.0%)	99 (72.8%)	87(64.0%)
0 (0%)	1 (1.0%)	3 (3.0%)	57 (64.8%)	94 (77.7%)	4 (4.0%)	32 (36.8%)
1 (1–10%)	14 (13.5%)	11 (10.9%)	13 (14.8%)	23 (19.0%)	14 (14.1%)	35 (40.2%)
2 (11–75%)	55 (52.9%)	56 (55.4%)	16 (18.2%)	4 (3.3%)	79 (79.8%)	19 (21.8%)
3 (>75%)	34 (32.7%)	31 (30.7%)	2 (2.3%)	0	2 (2.0%)	1 (1.1%)
ICC(2,1)	0.73	0.73	0.75	0.81	0.60	0.59
95% CI of ICC	0.64–0.80	0.63–0.80	0.66–0.81	0.74–0.86	0.48–0.70	0.43–0.67
ICC interpretation	moderate	moderate	good	good	moderate	moderate
No agreement	18 (13.2%)	12 (8.8%)	17 (12.5%)	4 (2.9%)	20 (14.7%)	19 (14.0%)
Agreement	118 (86.8%)	124 (91.2%)	119 (87.5%)	132 (97.1%)	116 (85.3%)	117 (86.0%)
0 or 1 (up to 10%)	16 (13.6%)	18 (14.5%)	97 (81.5%)	128 (97.0%)	24 (20.7%)	97 (82.9%)
2 or 3 (>10%)	102 (86.4%)	106 (85.5%)	22 (18.5%)	4 (3.0%)	92 (79.3%)	20 (17.1%)
ICC(2,1)	0.56	0.70	0.64	0.65	0.61	0.52
95% CI of ICC	0.43–0.67	0.60–078	0.53–0.73	0.55–0.74	0.49–0.71	0.39–0.73
ICC interpretation	moderate	moderate	moderate	moderate	moderate	moderate
No agreement	2 (1.5%)	8 (5.9%)	28 (20.6%)	12 (8.8%)	7 (5.1%)	12 (8.8%)
Agreement	134 (98.5%)	128 (94.1%)	108 (79.4%)	124 (91.2%)	129 (94.9%)	124 (91.2%)
Absent	1 (0.7%)	3 (2.3%)	57 (52.8%)	95 (76.6%)	4 (3.1%)	32 (25.8%)
Present (any)	133 (99.3%)	125 (97.7%)	51 (47.2%)	29 (23.4%)	125 (96.9%)	92 (74.2%)
ICC(2,1)	0.50	0.40	0.58	0.78	0.47	0.44
95% CI of ICC	0.40–0.61	0.25–0.53	0.45–0.68	0.70–0.84	0.33–0.59	0.30–0.57
ICC interpretation	moderate	poor	moderate	good	poor	poor

CI: confidence interval, ICC: intraclass correlation coefficient. Numbers in parentheses represent percentages of all cases in data rows 1 and 2 of all three parts of the table and of all cases with agreement in data rows below data row 2 of all three parts of the table.

**Table 4 cancers-16-01640-t004:** Individual and consensus classification of the histological infiltration patterns of ILC cases.

	Classical	Non-Classical (Together)	Solid	Tubular Elements	Trabecular	Alveolar
Any						
Observer 3	133	127	67	38	126	79
Observer 4	135	131	63	36	130	89
**Consensus**	**135**	**133**	**70**	**36**	**132**	**95**
11–100% (vs 0–10%)						
Observer 3	112	109	33	4	103	32
Observer 4	110	115	28	8	101	26
**Consensus**	**113**	**113**	**36**	**7**	**106**	**29**
>75%						
Observer 3	47 (9)	40	5 (1)	0	12	1
Observer 4	38 (5)	41	3 (1)	0	4	1
**Consensus**	**41 (3)**	**45**	**5 (1)**	**0**	**9**	**1**

Numbers in parentheses refer to pure pattern numbers, i.e., cases with 100% representation of the given pattern and no other components; zeros have been omitted.

**Table 5 cancers-16-01640-t005:** Multifocality and the presence of a mass lesion on the slides as evaluated by pathologists of the study.

	MULT(TOT)	MULT(ICCR)	Mass Lesion
No agreement	29	21	22
Agreement	107	115	114
Absent	35	99	20
Present	72	16	94
ICC(2,1)	0.52	0.52	0.43
95% CI of ICC	0.39–0.64	0.38–0.63	0.28–0.56
ICC interpretation	moderate	moderate	poor
Observer 3	90/136	20/136	106/136
Observer 4	84/136	33/136	105/136
**Consensus**	**92/136**	**30/136**	**108/136**

MULT refers to multifocality as defined earlier, MULT(TOT) is according to Tot [22], MULT(ICCR) is according to the International Collaboration on Cancer Reporting [24]. ICC: intraclass correlation coefficient; CI: confidence interval.

**Table 6 cancers-16-01640-t006:** A lack of mass lesions on mammography (A), histological slides (B), and both (C) and their associations with a predominant classical or non-classical pattern of ILC.

	Mass or Mass-like Lesion	No Mass or Mass-like Lesion	*p* Value *
A	Mass (on Mammogram)	No Mass (on Mammogram)	
Classical ILC (>75%)	21	18	
Non-classical ILC (>75%)	30	12	*p* = 0.114
B	Mass-like lesion (on slide)	No mass-like lesion (on slide)	
Classical ILC (>75%)	25	14	
Non-classical ILC (>75%)	38	5	***p* = 0.016**
C	Mass (on slide or mammogram)	No mass (on slide and mammogram)	
Classical ILC (>75%)	28	11	
Non-classical ILC (>75%)	39	4	***p* = 0.044**

* Fisher exact test. Note that there were mass-forming and non-mass-forming lesions (36 and 15, respectively, in part A; 43 and 7 in part B; 45 and 5 in part C) that could not be classified as of predominant (i.e., >75%) classical or non-classical patterns because they were rated as having between 10–75% of both without further specifications on dominant morphology.

**Table 7 cancers-16-01640-t007:** Associations between mammographic appearances and histological patterns.

>75% Histological Patterns/Hierarchical Mamographic Classification	Spiculated Mass	Lobulated Mass	Architectural Distortion/Increased Densitiy	Occult	All
Classical	20	1	15	3	39
Non-classical	30	1	10	2	43
Solid	3	1	0	0	4
Tubular elements	0	0	0	0	0
Trabecular	6	0	2	1	9
Alveolar	1	0	0	0	1

Note that case numbers of the individual non-classical patterns do not add up to equate the numbers in the cumulative non-classical patterns, as most cases had a mixture of non-classical patterns being present in <75% of the tumor.

## Data Availability

All data generated in the study are kept by the corresponding author and can be obtained upon reasonable request.

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
