# Peer review of "Histological Patterns and Mammographic Presentation of Invasive Lobular Carcinoma Show No Obvious Associations"

_cancers, 2024, doi:10.3390/cancers16091640_

Round 1

Reviewer 1 Report

Comments and Suggestions for Authors

The study entitled "Histological patterns and mammographic presentation of invasive lobular carcinoma show no obvious associations" performed by Cserni et al. shows the correlation between histological patterns and mammographic presentations of invasive lobular carcinoma (ILC).

Despite known mammographic features of ILCs like architectural distortion and occult lesions, spiculated mass was found to be the most common presentation.

Congratulations to the authors of the present study. I just have some comments:

- I suggest to modify the Title from "Histological patterns and mammographic presentation of invasive lobular carcinoma show no obvious associations" to "Histological patterns and mammographic presentation of invasive lobular carcinoma show no associations";

- The issue of different histological patterns is very critical in breast cancer management, since it can alter/influence subsequent prognosis and oncological outcomes. I suggest the authors to integrate this reference PMID: 38254865 to improve the quality of the manuscript;

- The Conclusions are too long. Do not include references in the Conclusions. Please modify it and make it 3-4 sentences max.

Author Response

The study entitled "Histological patterns and mammographic presentation of invasive lobular carcinoma show no obvious associations" performed by Cserni et al. shows the correlation between histological patterns and mammographic presentations of invasive lobular carcinoma (ILC).

Despite known mammographic features of ILCs like architectural distortion and occult lesions, spiculated mass was found to be the most common presentation.

Congratulations to the authors of the present study. I just have some comments:

RE: Thank you for your time and for the evaluation of our manuscript. Please see below our point-by-point reply, and the ensuing alterations of the manuscript.

- I suggest to modify the Title from "Histological patterns and mammographic presentation of invasive lobular carcinoma show no obvious associations" to "Histological patterns and mammographic presentation of invasive lobular carcinoma show no associations";

RE: Thank you for your suggestion. Your proposal to modify the title has been considered. There may still be some trends, e.g. classical and non-classical having a 3 to 2 ratio among architectural distortion/increased density and occult lesions; this was not a statistically significant difference but perhaps greater number of cases may reinforce these trends. To use an analogy, there is perhaps some difference in the shades of grey, but not a black and white, clear-cut association between the mammographic and the histologic appearance. In addition, based on the last comment of Reviewer 2, we found a weak association between classical pattern ILC and lack of mass lesions on slides (best match for a diffuse distribution). Therefore, we prefer to keep the word obvious in the title.

- The issue of different histological patterns is very critical in breast cancer management, since it can alter/influence subsequent prognosis and oncological outcomes. I suggest the authors to integrate this reference PMID: 38254865 to improve the quality of the manuscript;

RE: Thank you for drawing our attention to this publication, which contains valuable information that we will use in the future. The reference suggested deals with patterns of regressions (the scattered pattern with better prognosis vs the circumscribed) which do not fit the frames of the present study dealing with original histological patterns of lobular carcinomas (e.g. classical, solid, alveolar, trabecular, tubulolobular) of which cases post neoadjuvant therapy were excluded. This does not allow to document how the tumors we investigated would have behaved following neoadjuvant therapies, and how these histological patterns would affect regression patterns.

- The Conclusions are too long. Do not include references in the Conclusions. Please modify it and make it 3-4 sentences max.

RE: You are right. Generally, the Conclusions are much shorter. We have modified the manuscript accordingly. The thoughts of the conclusions (including the references) were moved to the end of the discussion, and a brief concluding section was added at the end of the manuscript text. This is just a few sentences long, and includes no references, and concentrates only on the main issues.

Reviewer 2 Report

Comments and Suggestions for Authors

The study by Cserni and colleagues is interesting. They studied cases formerly diagnosed as ILC and retrospectively re-analysed the mammographic features and histological classification of the tumour to determine whether imaging correlated with growth pattern. The important finding was that there was poor correlation between these 2 modes of diagnostics.

The authors also describe in some detail the challenges of classifying i) architectural patterns recognised in imaging, with generally a poor correlation seen between 2 observers; and ii) the histological variants of ILC on histopathology. Much of the paper is actually focused on these aspects of the study as opposed to the correlation between imaging and morphology. This is ok, as its important information for readers.

Comments:

Do you think 2 observers is sufficient for radiology review given the discordance observed? im not suggesting a 3rd reviewer is needed, but could this be described as a limitation or a prompt for >2 readers being needed to reach consensus?

Lines 250-260 – in this section you are talking about 'pure' patterns whether it refers to pure classical or pure non-classical types. I don’t think this data is in Table 4 but could it be added?

In Table 6 should the individual non classical types add up to the value given for the broad 'non-classical' row? ie there are 30 non-classical types with spiculated mass appearance, but only 10 defined as 1 of these individual types. Does this infer that the rest are mixed non-classical types? can you add that to the table or define whether this is the case? Also this table only shows those with >75% histology, so only 82 of the cases analysed. is it possible to incl more cases in this table so incl more histology parameters? or does that become too complicated to show? Noting that the abstract states '132 common cases were analyzed for possible associations between mammographic presentation and the histological patterns.’

Line 491-492: this is an interesting statement, in parentheses. please elaborate - is this 'diffuse' term referring to a radiological or morphological feature and is it published that this has a worse prognosis? is this referring to ILC or breast cancer in general too?

Author Response

The study by Cserni and colleagues is interesting. They studied cases formerly diagnosed as ILC and retrospectively re-analysed the mammographic features and histological classification of the tumour to determine whether imaging correlated with growth pattern. The important finding was that there was poor correlation between these 2 modes of diagnostics.

The authors also describe in some detail the challenges of classifying i) architectural patterns recognised in imaging, with generally a poor correlation seen between 2 observers; and ii) the histological variants of ILC on histopathology. Much of the paper is actually focused on these aspects of the study as opposed to the correlation between imaging and morphology. This is ok, as its important information for readers.

RE: Thank you for the time spent with our manuscript and for the comments aiming at improving it.

Comments:

Do you think 2 observers is sufficient for radiology review given the discordance observed? I’m not suggesting a 3rd reviewer is needed, but could this be described as a limitation or a prompt for >2 readers being needed to reach consensus?

RE: This is a well-placed comment, and you are right, this is a potential limitation. We think that with more observers, there can be a better estimation of reproducibility, as highlighted by ONEST analyses of other features (PMID: 32300181, 34415429, 34920295 and 36831541). These analyses can better estimate the minimum number of observers needed to evaluate such subjective tests for diagnostic consistency, and they can also highlight the minimum and maximum range of deviation between 2 observers. In fact, when there are only 2 observers, they could reflect the worst scenario (with maximum deviation from each other) or the best scenario (with the minimum deviation from each other) or anything in between, but without a formal ONEST type analysis, it cannot be decided what they represent from the whole spectrum of possibilities. However, the 2 observers reflect the double reading scenario implemented in breast cancer screening. (Although the 33/141 full agreement seems suboptimal, in each case this reflects the choice from all combinations of 5 potential radio-morphologies, on a yes or no basis, i.e. a single choice of 25=32 possibilities. The reproducibility of individual patterns was much better.) We have followed your advice and have inserted this as a limitation into the text of the manuscript with the following wording:

“Although the two-observer-setting reflects the double reading approach prescribed for mammography screening, it does not allow a proper evaluation of interobserver reproducibility. As a limitation of the present study, these agreement data reflect only the given two observers, but their degree of agreement reported can lie anywhere on the minimum and maximum scale that can be determined with ONEST (Observers Needed to Evaluate Subjective Tests) analyses using more observers [30-33]. The data must therefore be interpreted with this note of caution.”

Lines 250-260 – in this section you are talking about 'pure' patterns whether it refers to pure classical or pure non-classical types. I don’t think this data is in Table 4 but could it be added?

RE: This has been added to the table, and a note also defines what pure means in this context.

In Table 6 should the individual non classical types add up to the value given for the broad 'non-classical' row? ie there are 30 non-classical types with spiculated mass appearance, but only 10 defined as 1 of these individual types. Does this infer that the rest are mixed non-classical types? can you add that to the table or define whether this is the case? Also this table only shows those with >75% histology, so only 82 of the cases analysed. is it possible to incl more cases in this table so incl more histology parameters? or does that become too complicated to show? Noting that the abstract states '132 common cases were analyzed for possible associations between mammographic presentation and the histological patterns.’

RE: Your interpretation of the numerical mismatch in Table 6 (now renamed as 7) is correct: most cases were mixed, and this information has been added to the table as a note. Regarding the second half of the comment and the questions included, we faced the same problem while making the analysis and presenting the data. When the methods were devised, the results were not known, and it was and unexpected finding to have so few >75% pattern cases. This is the best approach to call an ILC as classical, alveolar, solid… etc. Since no tubule formation was noted in this extent, this pattern is missing from Table 6 (is represented by 0s). As the next category is that of 10-75% representation, this category does not allow for the classification of an ILC as this or that pattern, since the representation may be anything from 10% through less than 50% to more than 50% up to 75%. In retrospect, this would have been good to have a category for >50%, in order to allow for a dominant patter categorization, but owing to the amount of work this would require (several months were taken for sorting out old cases from the archives, looking at the slides again after long working hours...; and the cases have been filed back to the archives), we are not ready to provide this approach. Therefore, we had to add this limitation into the results as: “Only 82/132 cases (62%) could be assigned a histological pattern on the basis of >75% representation.”

Line 491-492: this is an interesting statement, in parentheses. please elaborate - is this 'diffuse' term referring to a radiological or morphological feature and is it published that this has a worse prognosis? is this referring to ILC or breast cancer in general too?

RE: This is based on published data. As mentioned in the introduction, diffuse breast cancers are a type of breast cancers that do not form masses on imaging and even on histological slides they are characterised by lack of masses and a spiderweb-like presentation (not the regular one of the epeirid, but something like this: https://www.euronews.com/2018/09/19/in-pictures-eerie-spiderweb-covers-entire-shoreline-in-greece). They have been associated with poor prognosis in ILCs, in the first description from 2003 (our reference 20). Later the approach of classifying breast cancers of any type according to their distribution (unifocal, multifocal or diffuse) was applied to 500 cases, and then an update on 2033 cases was also published (our references 22 and 23), and the prognostic impact (unifocal best, diffuse worst) was reinforced. The best evaluation of this subgross classification is with large section histology, which Prof. Tot uses on a regular basis, and we use on a selective ground. Although diffuse carcinomas are most commonly of lobular type, “ductal” (NST) carcinomas may also rarely show this distribution class (PMID: 23097710). Although diffuse cancers were not formally recorded in the study, some of the comments referred to their presence. This does not allow a statistical approach. However, your comment gave us the idea to check whether the lack of tumour mass on mammography or on the histological slides by gross assessment showed any association with dominant histological aspect. The lack of grossly identifiable tumour on the slides was associated with greater frequency of classical ILC! This has been elaborated, with additions to the results (including a new Table 6) and discussion. Thank you for this comment. We concentrated too much on the presence of features (mass, multifocality) rather than on the absence of something, which from the data we have would be the best approach to diffuse ILCs.